# Cultivating a Healthy Living Environment for Adolescents in the Post-COVID Era in Hong Kong: Exploring Youth Health Needs

**DOI:** 10.3390/ijerph19127072

**Published:** 2022-06-09

**Authors:** Cheuk-yeung Ho, Albert Lee

**Affiliations:** 1Centre for Health Education and Health Promotion, JC School of Public Health and Primary Care, The Chinese University of Hong Kong (CUHK), Hong Kong, China; ocean3398@gmail.com; 2Department of Rehabilitation Science, Hong Kong Polytechnic University, Hong Kong, China

**Keywords:** adolescent health, youth health risks, youth health needs, COVID-19, re-orientation of student health services

## Abstract

Studies have shown that adolescents now have a higher exposure to health risks than those in the past, and Hong Kong adolescents are no exception, particularly with the social crisis in 2019 and then the COVID-19 pandemic in 2020. Data from health care services for children and adolescents only represent the tip of the clinical iceberg, and health profiles, including living habits, lifestyles, data on health status, and health service utilization, are not always readily available for effective planning to cultivate a healthy living environment. In this paper, an exploratory study on secondary school students was conducted in one district of Hong Kong that has the potential and resources to become a healthy city. Students were found to have a high prevalence of poor sleep quality and quantity; excess screen time; physical inactivity; an insufficient intake of healthy food; emotional disturbance, including symptoms of post-traumatic stress disorders (PTSDs); and irregular attendance to student health services. A multivariate analysis showed that excessive time on social media websites, short sleeping hours, symptoms of PTSD, and a lack of regular exercise were independently associated with emotional distress. Youth service providers should re-orientate student health services, moving away from routine services to be more outreaching in order to cultivate a supportive living and learning environment, promoting better health for adolescents.

## 1. Background and Introduction

Prior to the COVID-19 pandemic, the level of well-being among students (school and college students) had already been a big concern, as we observed a high prevalence of youth risk behaviors, such as unhealthy eating, physical inactivity, substance misuse, and emotional disturbance, in many countries, both east and west [1,2,3,4,5]. Studies have shown that adolescents now have a higher exposure to health risks than those in the past. Therefore, adolescent health should be given higher priority in public health policies based on a systematic analysis of data reflecting the global health burden of adolescent heath [6,7] and the worldwide application of prevention science in adolescent health applications [8]. Accordingly, the 45th session of the UN Commission on Population and Development chose Adolescents and Youth as its central theme [9]. Since then, the COVID-19 outbreak has had a further significant impact on the mental health, education, and daily routine of students. Changes in daily routines, including a lack of outdoor activity, disturbed sleep patterns, and social distancing, have affected the mental well-being of students [10]. It has been observed that the decline in the level of physical activity and the prolonged use of electronic media have increasing effects on students’ learning, concentration, and sleep patterns (going to bed late and getting up late) [11]. Studies have also found an increasing prevalence of obesity [12,13] and myopia [14] among school children due to longer screen times; a lack of physical activity; and small, crowded living and learning spaces. A study on tertiary students found that over 60% experienced increased stress and around 30% experienced moderate-to-severe levels of depressive symptoms with the increasing use of electronic devices, and decreased participation in outside activities was positively associated with a higher level of depression severity [15]. Another study reported that 9% of students had depression and 14% had anxiety, and 25.4% stated that their mental health had deteriorated since the pandemic [16]. The health of adolescents has been further compromised by the COVID-19 pandemic.

Nine potentially protective aspects of youth and family behavior occurring during the prior month, including physical activity, time spent in nature and outdoors, appropriate screen time, and sleep quantity, have been identified in a longitudinal study to dampen the negative impacts caused by COVID-19 on adolescents’ well-being [17]. Adolescent health should place emphasis on the enhancement of the aforementioned protective factors and minimize risk factors. Nonetheless, the question remaining was “how should the services be delivered effectively to meet the needs of adolescents to reach the penultimate goal of health improvement?”

For the effective delivery of health services to children and adolescents, their voices should be heard, but, unfortunately, due to the circumstances of being a minor, they are usually only represented by adults, who may not necessarily reflect their best interests and real needs [18]. Evidence has highlighted the importance of including youth experience and voices in the planning, delivery, and evaluation of services [19]. In many countries, school health services play an active role in preventive care in the form of screening, preventive services, and health promotion, particularly regarding mental health [20]. The participatory approach involves children and adolescents and focuses on the conditions necessary to reduce risk factors, and collaborative services in primary health care can help to determine the needs of children and adolescents [20]. An assessment of their needs is required to reflect their best interests, and an analysis of their level of health risk behaviors is more relevant than morbidity and mortality data.

Adolescents do not usually perceive their risks of developing chronic conditions, as they happen years later, so they should be provided with data reflecting their immediate risks. Health communication targeting risk perceptions can lead to behavior change [21]. Studies have shown that interventions successfully changing risk perceptions often result in behavioral change [22,23]. Disease risk perception is a critical determinant of health behavior [24]. Adolescents’ risk perception can only be enhanced if they know the existence of the risks, and this may motivate them to expose themselves to protective factors. Studies are needed to investigate youth health behaviors during the COVID-19 pandemic in order to enable adolescents to perceive their risks, and service providers need to re-orientate their service provision in order to cultivate a supportive environment conducive to healthy living and to mitigate the negative repercussions caused by the COVID-19 pandemic with the maximum participation of adolescents. Youth service providers need to work upstream and address the determinants of health rather than downstream with the provision of routine services.

This study aims to provide a general overview of secondary school students’ health profiles, including health risk behaviors, lifestyle, and health status, and health-seeking behaviors, including the utilization of student health services (SHSs), during the COVID-19 pandemic. The findings of this study may fill the gaps in youth health needs assessments, as routine clinical data do not capture the data required to cultivate a better living environment for adolescents in the post-COVID-19 era, which is more important to protect and promote the health of adolescents and early identify potential risk factors.

## 2. Materials and Method

### 2.1. Study Design

This is a cross-sectional study, and it was carried out with the use of an electronic questionnaire to determine secondary school students’ school health profiles, including health risk behaviors, lifestyle factors, health status, and the utilization of student health services (SHSs). This study further analyzed the associations between health status and health risk behaviors, lifestyles, and SHS utilization.

The study was approved by the Survey and Behavioural Research Ethics Committee (SBRE) of the Chinese University of Hong Kong (SBRE-20-583). The survey was conducted anonymously. The participating schools obtained consent from parents and students, and students’ participation was entirely voluntarily, with no adverse repercussions.

### 2.2. Study Population

The study population was secondary school students (aged 11–18) in Kwai Tsing (K&T) District, one of the 18 districts in Hong Kong (HK), which has a district population of around 560,000 (about 8% of total HK population). It was chosen because the district is a pioneer in the Healthy City project [25], and it has established a framework for a medical–social–community model [26]. However, the median monthly income in K&T is among the lowest in Hong Kong. In 2019, the K&T District became the first district to establish a District Health Centre, which is part of a key government policy implemented to strengthen primary healthcare with the provision of nursing and allied health services, including social workers, and to support primary medical care providers in the district. The findings of this study may reflect the health needs of adolescents living in low socio-economic conditions and investigate how the community health initiatives help to meet those health needs.

### 2.3. Sample Population

Based on previous studies, the prevalence of students with emotional disturbance is around 25% [3], and for students performing regular exercise, the prevalence is also around 25% [4]. A sample size of 288 is needed for a margin of error of 5% (n = z^2^ × p(1 − p)/e^2^ z = 1.96 with a significance level at 5%, p = 0.25, and for population proportion and e = 0.05, the margin of error 1.96^2^ × 0.5 × 0.5/0.1^2^ = 288). Assuming π_0,_ the null hypothesis proportion is 0.25, the prevalence of certain health behaviors and health conditions is 25% for one group of students, and π = 0.35; for a higher proportion among another group of students, the sample will be 125 per group, giving a power of 80% (u = 0.84) of significance at 5% (v = 1.96) (N = [u √π (1 − π + v √π_0_ (1 − π_0_))/(π − π_0_)^2^ = [0.86 × 0.46 + 1.96 × 0.5]/0.15 × 0.15 = 125). Therefore, a sample of over 300 students was needed.

The K&T District is broadly divided into two main sub-districts, Kwai Ching and Tsing Yi. According to the 2016 Population By-Census, the total population and adolescent population (aged 10–19) are 336,405 and 32,355 in Kwai Ching, respectively, and 184,167 and 12,579 in Tsing Yi, respectively. Stratified cluster sampling was adopted to randomly select three secondary schools and two secondary schools from Kwan Chung and Tsing Yi, respectively, in order to reflect the proportion of the population of the two sub-districts. From each school, students from secondary 2 (S2), aged around 13–14, and students from secondary 5 (S5), aged around 17–18, were selected for this study to represent early and late adolescents. S1 students were new to the school, and S6 students were busy with public examinations, so S2 and S5 students were more appropriate study subjects.

### 2.4. Study Tools

This study utilized the Hong Kong Student Health Survey Questionnaire (HKSHQ) to measure health risk behaviors, lifestyle factors, and health status. HKSHQ is used by the Centre for Health Education and Health Promotion of the Chinese University of Hong Kong (CHEP) as a system of surveillance of student health status [27], and it is continuously refined as a tool for the assessment of student health status/health-related outcomes [11]. In a previous study, the tool was able to reveal that a decline in the level of physical activity and the prolonged use of electronic media had increasing effects on students’ learning, concentration, and sleep patterns (going to bed late and getting up late) [11]. Questions were added to report the utilization pattern of SHSs.

Questions screening for post-traumatic stress disorder (PTSD) were included, as the COVID-19 pandemic has had serious repercussions on mental health [16]. Before the outbreak of COVID-19, there was an intense social movement in 2019, and during the heights of the movement in late 2019, a public mental health crisis was generated, resulting in a prevalence of 11.2% for depression and 12.8% for probable post-traumatic stress disorder (PTSD) [28]. PTSD can be easily overlooked, as individuals may present with vague complaints with little or no account of the trauma that they experienced and emotional numbing. If PTSD is suspected, a few key screening questions may be useful to avoid problems in diagnosis, so this study adopted four key screening questions to alert professionals to the possibility of PTSD, which may require referral for further evaluation [29].

### 2.5. Data Collection

Letters of invitation were sent out to the schools sampled in March 2021, and they contained information about the study, a sample questionnaire, and a consent reply form. The schools were then provided a link so that students could access the electronic questionnaire and complete the questions.

### 2.6. Data Analysis

Descriptive data of all variables derived from the questionnaire were computed as categorical variables. Data analysis was performed using SPSS statistics software (version 25.0). The proportions of particular outcomes were tabulated. Certain categorical variables were converted into binary variables. The mode (most common) of sleep time duration reported by students was 6 h, so it was used as a dividing line. Regarding exercise, for substantial health benefits in adults, the Centre for Health Protection of Hong Kong Special Administrative Region (CHP) recommends engaging in at least 150–300 min of moderate-intensity aerobic physical activity; at least 75–150 min of vigorous-intensity aerobic physical activity; or an equivalent combination of moderate- and vigorous-intensity activity throughout the week [30]. Therefore, more than 60 min of moderate-to-vigorous intensity activity 3 days per week was used as a dividing line. CHP recommends 5 servings of fruit and vegetables [30], and the mode of fruit and vegetable intake reported in this study was a 0.5 serving, so 1 serving was used as a dividing line. For symptoms of PTSD, those with two symptoms were classified as positive to allow for a higher level of alertness [29]. For screen time, 2 h was chosen as a dividing line, as the American Academy of Pediatrics advises no more than 1–2 h of screen time for children [31].

The chi-square test was used to analyze differences among dichotomous dependent variables at a 5% level of significance. Multiple logistic regression was utilized to identify the independent variables associated with emotional distress (feeling sad and hopeless, and seeking help from others).

## 3. Results

A total of 585 students, comprising 120 and 159 male students and 134 and 172 females students from S2 and S5, respectively, participated in this study of five secondary schools randomly selected using stratified cluster sampling as described in the Section 2.

Table 1 shows the prevalence of different living habits (sleeping time and quality, and time spent on electronic media), lifestyles (diet and exercise), health status (emotional health and symptoms suggestive of PTSD), and the utilization of student health services. Around 30% of students slept less than 6 h per day with poor quality. Nearly half spent over 2 h watching television, playing electronic games, and using social media platforms. Most students had a very poor consumption of fruits and vegetables, with one serving or less, and a low level of regular exercise (Table 1). Over 20% of students felt emotional distress that affected their daily activities, and over 40% felt the need to seek help. Nearly 20% of students had symptoms suggestive of PTSD. However, nearly 80% of students did not attend student health services regularly. The prevalence of self-reporting substance misuse was found to be very low, at 0.2%.

Figure 1 shows the reasons for sleep disturbance. The four main reasons reported are failing to fall asleep within 30 min, nightmares, waking up in the middle of the night or early morning, and a hot environment. There is a significant association between sleep quality and sleeping hours, with 77.9% of students sleeping 6 h or more and only 48.5% of those sleeping less than 6 h rating their sleep quality as being good (*p* < 0.001). There is also a significant association between sleeping hours and time spent on social media, with 78.5% of students sleeping 6 h or more and only 63.1% of those spending less than 2 h on social media (*p* < 0.001). We observed a high proportion of students spending a lot of time using screens and reporting emotional disturbance, as can be seen in Table 1. This may be related to sleeping hours and quality of sleep.

The students’ level of exercise is far from the recommended level (Table 1). Figure 2 shows the reasons for not engaging in regular exercise. The main reasons reported are lack of time (42.6%), no interest (32.5%), academic stress (29.6%), and a lack of friends (19.3%).

Living habits and lifestyles may be correlated with screen time. Table 2 shows an analysis of the correlations between screen time (TV, electronic games, and social media) and sleeping time, vegetable intake, fruit intake, and exercise level. Only time spent on social media websites was found to correlate with sleeping time with statistical significance (78.5% of students spending less than 2 h on social media websites slept 6 h or more compared with 63.1% of those spending 2 h or more on social media, *p* < 0.001) (Table 2). Figure 3 shows the perceived impact of screen time on daily life and health. Eye discomfort, effects on studies, a lack of sleep, and a loss of concentration were reported as the major issues resulting from prolonged screen time.

Table 1 reveals that students’ emotional status is a cause of concern, with 22.4% feeling emotional distress that affects their daily activities, and 42.4% feeling sad and hopeless, and seeking help from others (Table 1). Table 3 shows the correlations between emotional status (feeling sad and hopeless, and seeking help from others) and living habits, lifestyles, screen time, and PTSD. The variables sleeping time (<6 h), low fruit intake (<1 portion), low level of regular exercise, excess time on social media websites, and symptoms of PTSD were found to have significant correlations with emotional status (Table 3).

Logistic regression was performed to identify the independent factors significantly associated with poor emotional status (Table 4). Sleeping less than 6 h, spending over 2 h on social media websites, and PTSD (2 or more symptoms) were independent variables found to have higher odd ratios of 1.62 (95 ci 1.08, 2.42), 1.77 (95 ci 1.23, 2.65), and 5.86 (95% ci 3.6, 9.6) of poor emotional status, respectively. Students with regular moderate or vigorous exercise (over 60 min over 3 days per week) had a lower odds ratio of 0.65 (95% ci 0.44, 0.95) of poor emotional status.

SHSs are a key source of preventive care for students in Hong Kong. With the high prevalence of unhealthy living habits and lifestyles, and poor emotional status, the low uptake rate of regular attendance to SHSs (77.3%, Table 1) should be explored. All primary and secondary students can enroll in SHSs to receive services that meet their health needs at various stages of their development, including physical examinations; screening for health problems related to growth, nutrition, psychological health, and behavior; and individual health counselling and health education. Figure 4 shows the reasons for not using SHSs regularly. Having a lack of time, being unfamiliar with the service, stressing over academic work, and being doubtful of the usefulness of the services are the main reasons for not using them regularly. To improve SHS attendance, 49.7% of students suggested that the services be organized within school hours, and 40.7% suggested that the services be organized within the school premises.

## 4. Discussion

This study found an inadequate sleeping time among adolescents. The American Academy of Sleep Medicine recommends that teens aged 13–18 years sleep 8–10 h per 24 h for optimal health [32]. The prevalence of short sleep duration (<8 h) among high school students in US national Youth Risk Behaviors Surveillance was found to be 72.7% [33]. Our study found that 28.3% of students slept less than 6 h and that 81.5% slept less than 8 h. Similar patterns were reported in other Asian countries, such as Taiwan [34] and Korea [35], and the average duration of sleep was 4.9 h in a Korean study [35]. The Adolescent Sleep Working Group reported that processes involved in regulating sleep timing seem to be altered to favor late nights across adolescent development [36]. The group also observed the trends of adolescents and young adults growing up in an electronic age and states that electronic exposure in the evening potentially disrupts sleep [37]. Our study found a high prevalence of media use amongst students. This was particularly the case during the COVID-19 pandemic as a result of school closures and with electronic media being the main channel of communication [11]. Engaging in a greater number and range of sleep-interfering activities, such as the use of electronic media before going to bed, has been shown to be associated with less nocturnal sleep and more daytime sleepiness in adolescents [38]. Our study showed that a higher proportion of students use social media websites less than 2 h daily with a sleeping time over 6 h with statistical significance (Table 2). Media use can cause increased sleep-disrupting mental, emotional, and physiologic arousal [39]. This might reflect the reasons for sleep disturbance, such as failing to fall asleep, nightmares, and waking up in the middle of the night or early morning (Figure 1), as well as the impact of excess screen time, such as eye discomfort, effects on studies, a lack of sleep, and a loss of concentration (Figure 3).

The prevalence of emotional disturbance was found to be high in this study, and it was found to be significantly higher among students with unhealthy living habits, including poor sleep, insufficient sleep time, and spending excess time on electronic media (Table 3). Sleeping less than 6 h and spending over 2 h on social media websites were found to have higher odd ratios of poor emotional status, which was analyzed using multiple logistic regression (Table 4). Self-reported sleep variables, such as trouble sleeping, tiredness, nightmares, and being a long sleeper, have been found to be significantly associated with psychological symptoms, including anxiety/depression and withdrawal [40].

Students who do not get the recommended amount of sleep for their age are at increased risk for chronic conditions, such as diabetes, obesity, and poor mental health, as well as injuries, attention and behavioral problems, and poor academic performance [32,37,41]. The quality of sleep is related to sleeping hours, as this study found that 51.5% of adolescents getting less than 6 h sleep reported poor sleeping quality compared with 22.1% of those getting 6 h or more sleep, and this was found to be statistically significant (*p* < 0.001). Sleep quality is as important as sleep duration in predicting future health, but it is often overlooked [42]. Studies have found very strong evidence tying sleep quality to the development of mental disorders, as it can more than double the risk of depression and anxiety [42].

A previous study has shown that adolescents spending more than 3 h per day on social media may be at heightened risk for mental health problems, particularly internalizing problems [43]. A poorer sleep quality can be a mediator on the pathway to internalizing problems [44]. Poor emotion regulation and a lack of social interaction may also be associated with social media use and contribute to symptoms of anxiety and depression [45]. However, adolescents may turn to social media as a form of escapism, perhaps to relieve emotional distress. The results of a longitudinal study revealed that increased time spent on social media was not associated with increased mental health issues across development when examined at the individual level [46]. Emotional problems are multifactorial, involving person-centered characteristics, such as biological predispositions; coping mechanisms; and experiencing certain situations, such as traumatic events and prolonged exposure to stress; and sleep quality.

Time spent on social media is an important factor associated with mental health problems, as the study conducted by Riehm at al. found that, if adolescents who used social media for more than 30 min per day instead used it for 30 min or less, there would have been 9.4% fewer high internalizing problem cases and 7.3% fewer high externalizing problem cases [43]. However, the findings from the study conducted by Coyne et al. suggest also examining the context and content surrounding social media use and the many other factors that might explain the increase in mental or emotional health problems during adolescence [46]. In this study, we examined other factors, such as living habits, lifestyles, and symptoms of PTSD, as independent variables.

A major mental health burden was identified during the COVID-19 pandemic [16] and social unrest in 2019 [28]. Post-traumatic growth (PTG), defined as the positive psychological changes that happen upon coming to terms with highly stressful events, has been found to more likely occur among people with higher post-traumatic stress [47]. Students were profoundly impacted by the social unrest in the second half of 2019 [28], and then they faced the COVID-19 pandemic, so the odds ratio of poor emotional status was found to be very high (OR 5.86, 95% ci 3.6, 9.6, Table 4) with symptoms of PTSD. Screening for PTSD symptoms would enable the early detection of those at risk. Patient education and social support are important initial interventions to engage the patient and mitigate the impact of the traumatic event, and the rapid engagement of treatment, early and ongoing social support, and the avoidance of retraumatization have been shown to be associated with a good prognosis [48].

A systematic review found evidence of a significant, cross-sectional relationship between unhealthy dietary patterns and poorer mental health in children and adolescents [49]. In this study, in the univariate analysis, a higher proportion of students with a poor intake of fruit were found to feel sad and hopeless (Table 3). The association was not found in the multiple regression analysis. Overall, there was a very high prevalence of a poor intake of fruit and vegetable among students (over half had less than one portion a day, Table 1). Regular exercise was found to be associated with lower odds of having emotional problems with statistical significance in the univariate and multi-variate analyses (Table 3 and Table 4). Physical activity is well recognized as a key factor in the prevention and management of mental illness, including mental disorders, such as depression and anxiety, as well as the promotion of mental health, such as well-being [50]. Physical activity was also found to be a good and effective choice to mitigate the negative effects of the COVID-19 pandemic on mental health during the first year of the COVID-19 pandemic [51].

A longitudinal study conducted by Rosen found an association between low passive screen time use, low news media consumption about the pandemic, a structured daily routine, spending more time in nature, and getting the recommended amount of sleep with better mental health outcomes in youth during the pandemic [17]. Encouraging more physical activity can also help adolescents to spend more time in nature. Appropriate screen time can improve sleep hygiene and can also have direct effects on mental health. Cultivating positive lifestyles and living habits, providing a physical environment conducive to regular exercise, and providing a supportive social environment to help adolescents coping with post-traumatic stress can enhance their emotional well-being in the post-COVID-19 era. The home and school environment should empower adolescents to adopt a more structured daily routine, regularly exercise, and access resources for emotional support. The Adolescent Sleep Working Group has reported evidence that strongly implicates earlier school start times (i.e., before 8:30 a.m.) as a key modifiable contributor to insufficient sleep, as well as circadian rhythm disruption [36]. The starting time for school should be further explored to enable students to gain sufficient sleep, both in quantity and quality.

The mental health burden among adolescents is a serious concern, and it is expected to further increase with the COVID-19 pandemic. The low level of regular utilization of SHSs will not enable the service to play an active role in preventive care, particularly health promotion, including mental health [20]. The students’ lack of participation cannot allow the services to focus on the necessary conditions to reduce risk factors and determine the needs of children and adolescents [20].

There are limitations to this study. Self-reporting might not capture all the data. However, it is the best option during the COVID-19 pandemic, and it also mobilizes the co-operation of youth; a cross-sectional study cannot investigate causal relationships. The findings of associations may still be useful for the planning of a future longitudinal study. This study is only based on one district in Hong Kong. The K&T district was chosen for a particular reason. The district is a pioneer in the healthy city movement, and it is also the first district with a government-commissioned District Health Centre. The findings can be a useful reference to help the district to strengthen its primary healthcare services in order to target the needs of adolescents. With the existing experience and culture of the healthy city movement, together with the infrastructure of the District Health Centre, the K&T district has the potential to develop a new model of comprehensive adolescent care that is user friendly and promotes the health of adolescents in the context of daily living.

## 5. Conclusions

The findings of this study reflect the unmet needs of students and provide some insights into the types of services that serve their best interests. Service providers for adolescents should think outside the box to be more outreaching in order to promote a better living environment for the maintenance of healthy behaviors. A systemic review identified the key components of user-friendly youth services [52]. These include accessibility; staff being supportive, respectful, and trustworthy, actively listening and providing clear information; perceived medical competency; the assurance of privacy, confidentiality, and patient autonomy; comprehensive and continuity of care; youth involvement in healthcare; an age-appropriate environment; and health outcomes focusing on quality of life. The services should be based in the locality where the adolescents live and study. The emphasis should be put on a paradigm shift from adolescent health services to adolescent health promotion, and the approach should be setting-based health promotion (healthy setting) involving the population in the context of their everyday life, combining complimentary approaches that include the school setting, local community, and local primary care providers [53,54]. The “healthy setting” approach allows us to explore the influences of inter-relationships and interactions between people, places, programs, and policies on health, cultivating a living environment conducive to positive youth health [53,54]. This is particularly important for mental health, as primary and secondary preventions are far more cost effective than the treatment of mental disorders (tertiary prevention). Local family doctors and the local District Health Centre are in unique positions to work and engage the key stakeholders, adolescents, families, schools, and other local youth services to set this new paradigm.

## Figures and Tables

**Figure 1 ijerph-19-07072-f001:**
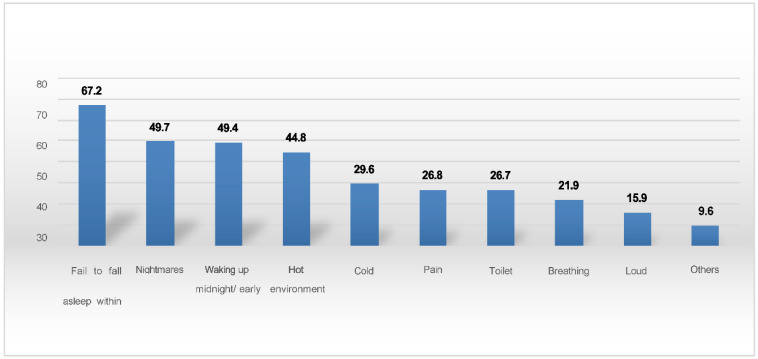
Reasons for sleep disturbances.

**Figure 2 ijerph-19-07072-f002:**
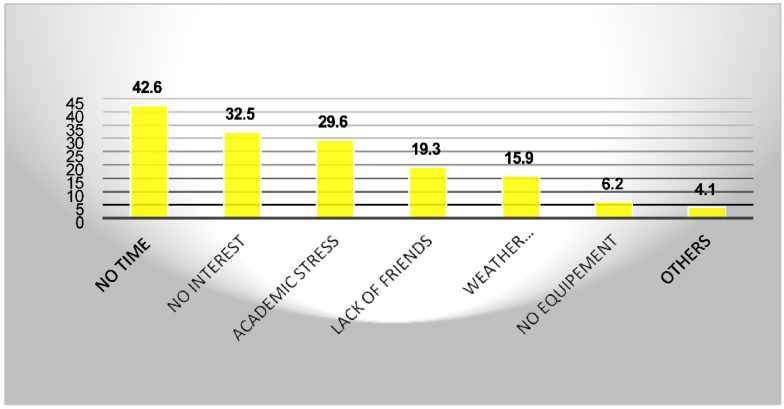
Reasons for not exercising.

**Figure 3 ijerph-19-07072-f003:**
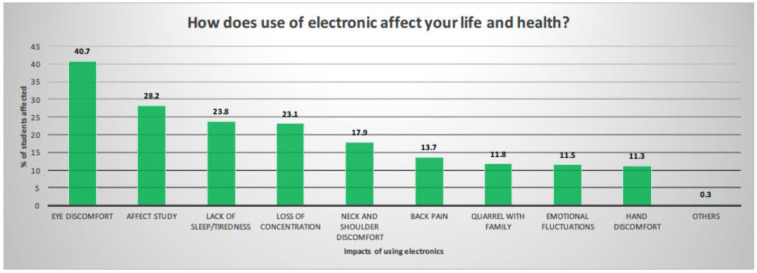
Perception of use of electronic media on health and daily life.

**Figure 4 ijerph-19-07072-f004:**
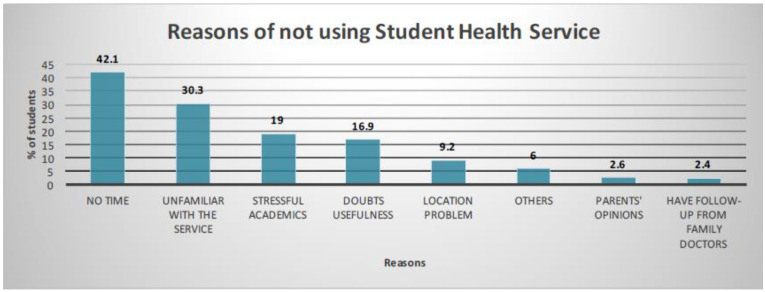
Reasons for not utilising Student Health Services.

**Table 1 ijerph-19-07072-t001:** Prevalence of living habits, lifestyles, and health status (N = number of students responding to the question with a valid answer).

Living Habits, Lifestyles, and Health Status	Prevalence Number of Students in Brackets	N (Total Number of Students Who Responded)
Sleeping < 6 h daily	28.3% (151)	533
Poor quality of sleep	33% (176)	533
Watching TV > 2 h daily	55% (293)	533
Electronic games > 2 h daily	56.7 (302)	533
Time on social media websites > 2 h daily	42.5% (227)	533
<1 serving of vegetables daily	65.3% (308)	472
<1 portion of fruit daily	55.9% (327)	585
60 min of regular moderate or vigorous exercise 3 days or more per week	31.9% (170)	533
Not having breakfast everyday	61.5% (328)	533
Irregular attendance to student health services	77.3% (412)	533
Feeling sad and hopeless, and seeking help from others	42.4% (226)	533
Emotional distress affecting usual activities	22.4% (131)	585
Self-harm	8.9% (47)	533
Suicide plan	6.4% (34)	533
Substance misuse (taking medication not for medical purposes)	0.2% (9)	585
PTSD (2 symptoms or more)	19.5% (114)	585

**Table 2 ijerph-19-07072-t002:** Correlations between time spent on electronic media and sleeping time, vegetable intake, fruit intake, and exercise level.

	Sleep ≥ 6 h	Sleep < 6 h	*p*-Value	Vegetable < 1 Bowl	Vegetable ≥ 1 Bowl	Not Sure	*p*-Value	Fruit < 2 Servings	Fruit ≥ 2 Servings	Not Sure	*p*-Value	Exercise < 5 Days	Exercise ≥ 5 Days	*p*-Value
TV ≥ 2 h	70.0%	30.0%	0.292	57.3%	32.1%	10.6%	0.944	86.7%	7.2%	6.1%	0.324	86.7	13.3	0.797
TV < 2 h	73.9%	26.1%		56.2%	32.5%	11.3%		82.2%	9.6%	8.2%		86	14	
Electronic games ≥ 2 h	69.3%	30.7%	0.103	59.6%	28.9%	11.4%	0.133	86.1%	7.5%	6.3%	0.428	86.4	13.6	0.922
Electronic games < 2 h	75.4%	24.6%		53.0%	36.8%	10.3%		82.2%	9.5%	8.3%		86.2	13.8	
Social media ≥ 2 h	63.1%	36.9%	<0.001 ***	56.2%	32.9%	10.8%	0.962	87.1%	6.4%	6.4%	0.261	83.5	16.5	0.091
Social media < 2 h	78.5%	21.5%		57.1%	31.8%	11.0%		82.4%	9.8%	7.7%		88.4	11.6	

*** *p* < 0.001.

**Table 3 ijerph-19-07072-t003:** Correlations between emotional status and living habits, lifestyles, and health status.

	Feeling Sad and Hopeless and Seeking Help from Others		N	*p* Value
	Yes	No		
Parental education level both at secondary or above	61.2%	73.95	428	0.33
On CSSA	14.8%	21.3%	426	0.17
Sleeping < 6 h	38.2%	21.1%	584	<0.001 ***
Sleep quality Poor	39.6%	22.85	580	<0.001 ***
Vegetable intake < 1 bowl	64.7%	62%	521	0.31
Fruit intake < 1 portion	68.4%	63%	545	0.02 *
Not having breakfast everyday	65.1%	59%	585	0.17
Regular moderate or vigorous exercise 3 days or more in a week	26.4%	36.7%	585	0.01 **
Watching TV > 2 h per day	51%	50%	555	0.74
Electronic games > 2 h per day	43.8%	42.5%	585	0.8
Spending time on social media websites > 2 h per day	51.3%	35.5%	585	<0.001 ***
Irregular use of student health services	77.8%	75%	585	0.49
PTSD (2 or more symptoms)	34.5%	7.4%	585	<0.001

* *p* value < 0.05. ** *p* value < 0.01. *** *p* value < 0.001.

**Table 4 ijerph-19-07072-t004:** Lifestyles and health status were found to be significantly associated with emotional status, feeling sad and hopeless, and seeking help from others, using multivariate analysis.

	Odds Ratio	95% Confidence Interval	*p* Value
Sleep < 6 h	1.62	1.08, 2.42	0.018
Regular moderate or vigorous exercise 3 days or more in a week	0.65	0.44, 0.95	0.027
Spending > 2 h on social media websites	1.77	1.23, 2.65	0.02
PTSD (2 or more symptoms)	5.86	3.6, 9.6	<0.001

## Data Availability

Not applicable.

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
