# Peer review of "Cultivating a Healthy Living Environment for Adolescents in the Post-COVID Era in Hong Kong: Exploring Youth Health Needs"

_ijerph, 2022, doi:10.3390/ijerph19127072_

Round 1
Reviewer 1 Report
In this work, an exploratory study on secondary school students was conducted in one district of Hong Kong with potential and resources to become a healthy city. Students were found to have high prevalence of poor sleep quality and quantity, excess screen time, physical inactivity, insufficient intake of healthy diet, emotional disturbance including symptoms of post-traumatic stress disorders (PTSD). Some comments are listed as follows,
1. I suggest to further refine the abstract to highlight the significance and innovation of the presented work.
2. Many studies are cited without stating what are the main conclusions and how they relate to the specific work carried out in the manuscript.
3. There is no clear statement describing what are the research gaps that the paper is trying to fill in.
4. The study tools used in this work would be better verified. Even data from reference would also be great.
5. This paper lacks innovations and is of low quality. Although many data and curves are shown, they are not well organized in an informative way. Meanwhile, most of the major conclusions are easy to think about.
Reviewer 2 Report
Dear Authors,
Thank you for the opportunity to review such an interesting manuscript. The overall concept of the manuscript is good. The study was conducted reliably and the results are generally presented in a clear manner, the quality of the figures should be improved (legibility, descriptions).
Comments are given in the attached file.
Best regards,
Reviewer

Reviewer 3 Report
The article analyzes a very relevant topic that is important for countries around the world, for their populations because it affects teenagers, our future generation.
A representative study showed risk factors, their importance to adolescents for their mental health, sleep duration, lack of exercise, too much time off screen, physical inactivity, malnutrition, symptoms of emotional disorders, including post-traumatic stress disorder (PTSD), irregular attendance at student health maintenance services.
The article is well illustrated, the diagrams are understandable and clear. With so much material, more specific conclusions would be needed on what measures to take to improve adolescent mental health, what measures are needed first, and if the authors are unable to write this, it should explain what else needs to be done, what to research. Are the results of the teen survey in Hong Kong applicable in other countries? If so, in which ones.?
Round 2
Reviewer 1 Report
The authors have replied my comments properly.